# The Relevance of Physico-Chemical Properties and Protein Corona for Evaluation of Nanoparticles Immunotoxicity—In Vitro Correlation Analysis on THP-1 Macrophages

**DOI:** 10.3390/ijms23116197

**Published:** 2022-05-31

**Authors:** Mojca Pavlin, Jasna Lojk, Klemen Strojan, Iva Hafner-Bratkovič, Roman Jerala, Adrijana Leonardi, Igor Križaj, Nataša Drnovšek, Saša Novak, Peter Veranič, Vladimir Boštjan Bregar

**Affiliations:** 1Institute of Biophysics, Faculty of Medicine, University of Ljubljana, Vrazov trg 2, SI-1000 Ljubljana, Slovenia; 2Group for Nano and Biotechnological Application, Faculty of Electrical Engineering, University of Ljubljana, Tržaška 25, SI-1000 Ljubljana, Slovenia; jasna.lojk@ffa.uni-lj.si (J.L.); klemen.strojan@gmail.com (K.S.); vladimir.bregar@gmail.com (V.B.B.); 3Department of Synthetic Biology and Immunology, National Institute of Chemistry, Hajdrihova 19, SI-1000 Ljubljana, Slovenia; iva.hafner@ki.si (I.H.-B.); roman.jerala@ki.si (R.J.); 4EN-FIST Centre of Excellence, Trg Osvobodilne fronte 13, SI-1000 Ljubljana, Slovenia; 5Department of Molecular and Biomedical Sciences, Jožef Stefan Institute, Jamova 39, SI-1000 Ljubljana, Slovenia; adrijana.leonardi@ijs.si (A.L.); igor.krizaj@ijs.si (I.K.); 6Department for Nanostructured Materials, Jožef Stefan Institute, Jamova 39, SI-1000 Ljubljana, Slovenia; natasa.drnovsek@ijs.si (N.D.); sasa.novak@ijs.si (S.N.); 7Institute of Cell Biology, Faculty of Medicine, University of Ljubljana, Vrazov trg 2, SI-1000 Ljubljana, Slovenia; peter.veranic@mf.uni-lj.si

**Keywords:** nanoparticles, nanotoxicology, immune response, correlation, protein corona, cytokines, inflammasome, macrophages, physico-chemical properties, TiO_2_

## Abstract

Alongside physiochemical properties (PCP), it has been suggested that the protein corona of nanoparticles (NPs) plays a crucial role in the response of immune cells to NPs. However, due to the great variety of NPs, target cells, and exposure protocols, there is still no clear relationship between PCP, protein corona composition, and the immunotoxicity of NPs. In this study, we correlated PCP and the protein corona composition of NPs to the THP-1 macrophage response, focusing on selected toxicological endpoints: cell viability, reactive oxygen species (ROS), and cytokine secretion. We analyzed seven commonly used engineered NPs (SiO_2_, silver, and TiO_2_) and magnetic NPs. We show that with the exception of silver NPs, all of the tested TiO_2_ types and SiO_2_ exhibited moderate toxicities and a transient inflammatory response that was observed as an increase in ROS, IL-8, and/or IL-1β cytokine secretion. We observed a strong correlation between the size of the NPs in media and IL-1β secretion. The induction of IL-1β secretion was completely blunted in NLR family pyrin domain containing 3 (NLRP3) knockout THP-1 cells, indicating activation of the inflammasome. The correlations analysis also implicated the association of specific NP corona proteins with the induction of cytokine secretion. This study provides new insights toward a better understanding of the relationships between PCP, protein corona, and the inflammatory response of macrophages for different engineered NPs, to which we are exposed on a daily basis.

## 1. Introduction

In the last few decades, nanotechnology has emerged as a promising new field, and several research breakthroughs have been implemented in industrial and medical products. Various nanoparticles (NPs) can be found in food processing (food-grade SiO_2_, TiO_2_, and iron oxide) [1,2,3,4], cosmetics (TiO_2_, silver, Au, and ZnO) [5], coatings, (silver, TiO_2_, and SiO_2_, etc.) and biotechnology [6]. Moreover, several NP formulations have been approved for biomedical applications [7] and as vehicles for drug delivery (e.g., Cliavist, Neulasta, Depodur, and Abraxane) [8]. Exposure to these NPs mainly occurs through different biological tissue barriers, such as the gastrointestinal tract (GIT), lung epithelium, or skin, which makes it difficult to evaluate the exposure dose, especially for NPs that are used in consumer products. It has been estimated that an average adult consumes a daily dose of 0.2–3 mg kg^−1^ of food-grade TiO_2_ (E171) [9,10], while children in USA consume as much as 5–10 mg kg^−1^ per day [11]. Even more concerning are recent studies that link the exposure and long-term accumulation of NPs to the emergence of certain diseases, such as diabetes, where TiO_2_ NPs were found in the pancreatic tissue of diabetic patients [12].

Despite the increased exposure, there is still insufficient data available for identifying general rules that determine the nanotoxicity and immunogenicity of certain NP formulations [13]. Various studies have analyzed the interaction of nanoformulations with the innate immune system [14,15,16,17,18,19,20,21,22,23,24,25,26,27], including complement system activation [16,17,24,28], cytokine expression [22,29], and different responses of macrophages [14,18,21,25,26,28]. It was also shown that various types of commonly used NPs such as TiO_2_ or SiO_2_ can cause NLR pyrin domain containing 3 (NLRP3) inflammasome activation [18,30,31,32,33,34,35,36]. However, the evaluation of potential nanotoxicity still presents a large and complex problem due to lack of standardization [4,37,38,39,40,41], appropriate models, a vast number of different NP types, and variations in analysis and exposure protocols, as well as a poor understanding of interactions between NPs and immune cells. Recent documents from the EFSA Panel on Food Additives and Nutrient Sources have stated that there are inconclusive results on the potential toxicity and immunogenicity of several very common engineered NPs, such as SiO_2_ and TiO_2_ [3,11,40], and several review papers have stressed that there is a lack of studies that would analyze physiochemical properties (PCP) and protein corona composition in relation to immunotoxicity [13,38,42,43,44,45,46]. 

In the last few years, computational nanotoxicology, and in particular, nano-QSAR (quantitative structural–activity relationships) were developed [47,48,49] in order to analyze the relationships between the physico-chemical properties and the activity of NPs. Nonetheless, studies that quantitatively analyze together PCP, protein corona, and immunogenicity are scarce, and knowledge regarding different factors that govern toxicity and immunogenicity of NPs is still emerging [4,13,41,50].

In this work, we focused on the relationship between PCP, protein corona composition, and the biological response of THP-1 human macrophages to different NPs, in order to identify the parameters that determine cytotoxicity and immunogenicity. We have selected seven engineered NPs that are commonly used in various consumer products (P25 industrial-grade TiO_2_, food-grade TiO_2_, cosmetic-grade TiO_2_, SiO_2_, silver, and polyacrylic acid (PAA)-coated magnetic NPs). We determined the hydrodynamic diameter, Z-average, zeta potential, and protein corona composition using mass spectrometry (MS), and leached ions using ICP-MS analysis. Toxicity and immunotoxicity were measured using different endpoints: viability, membrane damage, ROS, and cytokine secretion. Correlations between measured parameters were analyzed via the calculation of Spearman’s coefficient. We observed a proinflammatory response in terms of IL-8 and IL-1β cytokine secretion at moderate NP concentrations for SiO_2_ and several TiO_2_ NPs. Importantly, cytokine secretion strongly correlated with the size of the NPs in the culture media, and the presence of some specific proteins in the protein corona correlated with cytokine secretion. Although only a limited number of different NPs were evaluated, this study provides new insights and a quantitative analysis into the relations between NP properties, protein corona composition, and cell responses for different engineered NPs to which we are exposed on a daily basis through food, cosmetics, and materials.

## 2. Results

The aim of this study was to analyze and to correlate the physicochemical properties of NPs and the protein corona composition with the responses of THP-1 human macrophages in vitro. For the analysis, we selected eight types of engineered NPs: (a) biomedical magnetic PAA-coated cobalt–ferrite NPs used for labelling and separation [51,52], (b) engineered SiO_2_ used in cleaning materials [53], (c) silver NPs as NPs that are widely used in various materials [54], (d) P25 [2,55,56,57,58], (e) ANATAZ TiO_2_, a reference TiO_2_ material [2], (f) Dr. Adorable (DrA) TiO_2_ [2], (g) Naturally Balmy (N.B.) TiO_2_ NPs used in cosmetics [2], and (h) food-grade (FG) TiO_2_ that is used in food industry products.

### 2.1. Characterization

Characterization was performed in the distilled water in which the stock solutions of NPs were prepared, and in complete cell culture medium RPMI-1640 with 10% (*v*/*v*) FBS (RPMI). The results of PCP are presented in Figure 1. The hydrodynamic diameter (peak of the number distribution) of NPs ranged from 21 nm for silver NPs to 1854 nm for N.B. TiO_2_ in water, and from 4.5 nm for SiO_2_ NPs to 1374 nm for DrA TiO_2_ in RPMI (Figure 1A and Appendix A). The differences between the NPs were also observed with transmission electron microscopy, where micrographs of NPs dispersed in water showed large aggregates for N.B., P25, and ANATAZ TiO_2_, and single particles for silver NPs (Appendix A).

SiO_2_ NPs were already aggregated in water, which was reflected in the large Z-average (Figure 1B) and PDI values (Figure 1C), and the sedimentation of the suspension, which led to an underestimated hydrodynamic diameter. There were large differences between the various types of TiO_2_ NPs, in agreement with previously published data [2], where different compositions of powder surfaces were suggested as the reason for the observed differences in zeta potential and particle size. Differences in zeta potential between various NPs (Figure 1D) were much larger in water than in RPMI. In water, P25, N.B., and ANATAZ TiO_2_ were all positively charged, while PAA, silver, and FG TiO_2_ were all negatively charged. In RPMI media with 10% FBS, all NPs had a negative zeta potential ranging between −14 mV and −6.6 mV. The zeta potential measured in complete cell culture media tended to be negative, due to the formation of the protein corona on the surface of the NPs and the presence of free proteins in the sample [51,53,59]. 

### 2.2. Impurities

Impurities are a common but often-overlooked component of different nanomaterials. They originate either from the material used for NP preparation, or from other components that are used in the manufacturing process. Importantly, these impurities can be responsible for the activation or suppression of the immune system, and they can consequently lead to artifacts in immunotoxicity studies [44,60,61,62,63]. Therefore, the LAL test was performed to confirm that there was no endotoxin contamination. To evaluate the leachability of metal ions from the NP suspensions, the samples were aged in physiological NaCl solution for 24 h, and the concentration of elements was measured using ICP-MS (Figure 2). In general, Si was found in higher quantities than other elements, due to sample preparation in glass bottles. As expected, silver NPs leached substantial quantities of silver ions. We also detected the leaching of the impurities Zn, Mg, and Fe from DrA TiO_2_ NPs, and Mg and Fe from silver NPs; Mg was also detected in ANATAZ and N.B. TiO_2_. As expected for PAA coated cobalt–ferrite NPs, the leaching of Co was detected, while for SiO_2_ NPs, substantial amounts of Si were present. 

### 2.3. NP Corona Protein Composition

The composition of the NP protein corona after the incubation of NPs in complete cell culture medium RPMI-1640 with 10% (*v*/*v*) FBS was analyzed using MS (Figure 3). A complete list of identified proteins, their theoretical isoelectric points (pIs), molecular masses, and biological functions can be found in the Appendix A, and the relative percentages of proteins with a relative abundance in protein corona that is greater than or equal to 5%, in Appendix A. Differences in the relative protein abundance in coronas between different NPs can be clearly observed. The largest numbers of different proteins were found in the protein coronas of SiO_2_, PAA, and P25 TiO_2_ NPs. In the analysis, 28 different proteins were identified, out of which only 15 proteins were detected in at least two independent replicates of the same NPs, and these are presented in Figure 3 and Appendix A. Six proteins were common to all NPs. As expected, serum albumin (ALBU), which is the main component of FBS, was also the most abundant protein in all NPs coronas and in the negative control sample. Other proteins that were present on all NPs were α-1-antiproteinase (A1AT), apolipoprotein A-I (APOA1), α-2-HS-glycoprotein (FETUA), hemoglobin subunit α (HBA), and hemoglobin fetal subunit β (HBBF), out of which, A1At and FETUA were also present in the negative control. 

The majority of detected proteins, e.g., serum albumin (ALBU), apolipoprotein A-I (APOA1), and hemoglobin fetal subunit β (HBBF), are involved in molecular transport processes (Appendix A). The other most frequent biological process in which the identified corona proteins are involved is the immune response. Complement factor H (CFAH) was found in P25 TiO_2_, and N.B TiO_2_. In the coronas of other NPs, no proteins of the complement system were detected. The most commonly detected corona proteins, including albumin, are negatively charged at physiological pH (Appendix A), and positively charged proteins were only present in small amounts. 

### 2.4. Viability

Cell viability was determined with the Hoechst/PI differential staining assay following 24 h incubation with increasing concentrations of NPs. The lower concentrations (2 and 10 µg mL^−1^) were chosen for analyzing the effects of biologically relevant concentrations, and higher concentrations (50 and 100 µg mL^−1^) were chosen for an easier comparison of our results with the literature and the analysis of more subtle NP-induced effects. As seen in Figure 4, increasing the NP concentration consistently reduced the cell number, although for certain NPs such as PAA, ANATAZ TiO_2_, FG TiO_2_, and N.B. With TiO_2_, the effect was low and there was no increase in the dead, PI-positive cells. The highest effect on cell viability was induced by silver NPs, which resulted in a 0% viability at a concentration of 100 µg mL^−1^ (results not shown). A prominent decrease in viable cells and a corresponding increase in dead cells was also observed in SiO_2_, DrA TiO_2_, and P25 TiO_2_ NPs. Differentiated THP-1 cells do not proliferate, and so the observed effects were a sole effect of direct toxicity and not a result of altered cell division.

### 2.5. Nanoparticle Internalization

To confirm NP internalization, TEM was performed following 24 h incubation with 50 µg mL^−1^ NPs (25 µg mL^−1^ for silver NPs). Internalization was confirmed for all NPs (Figure 5). In all cases, NPs were found to be enclosed in different vesicles of the endolysosomal pathway (early endosomes, late endosomes, lysosomes, amphisomes), indicating that the NPs do not interfere with vesicle maturation and intracellular trafficking. No NPs were found in the cytosol or and none were associated with any other organelles. Except for ANATAZ TiO_2_, NPs were mostly found in loose aggregates inside the vesicles. Clear differences could also be observed in the structures and sizes of the different TiO_2_; the N.B. and P25 TiO_2_ aggregates had a size distribution ranging from 50 to 100 nm; the DrA and FG TiO_2_ formed aggregates over a range of 100–300 nm; and ANATAZ TiO_2_ had the largest and most densely packed aggregates (>600 nm). On the other hand, PAA and silver NPs were very stable and 20 nm in size. Interestingly, while SiO_2_ formed large aggregates, only the smaller SiO_2_ aggregates were internalized. It should be noted that with TEM, we can see the crystal core but not the organic layer (PAA polymers or the protein corona that is formed around NPs or between NP-forming aggregates). 

### 2.6. Reactive Oxygen Species

To determine if NPs induce oxidative stress in THP-1 cells, ROS were determined with a CM-H_2_DCFH-DA assay following 24 h incubation with increasing concentrations of NPs (Figure 6). PAA, FG TiO_2_, and ANATAZ TiO_2_ did not induce ROS, while for silver NPs, a small decrease in ROS could be observed. Other NP types induced a concentration-dependent increase in ROS, with P25 TiO_2_ leading to an almost 4-fold increase at a 100 µg mL^−1^ concentration. No changes in ROS were detected for the lowest, more physiologically relevant concentration (10 µg mL^−1^) of any NP type. 

### 2.7. Cytokine Secretion

To determine if NPs can activate THP-1 cells and induce cytokine secretion, we analyzed the secretion of Interleukin 6 (IL-6), IL-8, and tumor necrosis factor α (TNF-α) following 24 h incubation with increasing concentration of NPs (Figure 7, Appendix A). There was no increase in IL-6 and TNF-α secretion (Appendix A). On the other hand, P25 ANATAZ, DrA, and FG TiO_2_ induced a concentration-dependent secretion of IL-8. 

### 2.8. IL-1β Secretion and NLRP3 Inflammasome Activation

NPs aggregates can induce NLRP3 activation through several possible mechanisms including ROS and lysosome destabilization [64,65,66]. Inflammasome activation leads to proinflammatory cytokine IL-1β maturation and secretion, which we used as a readout of inflammasome activation. As pro-IL-1β and also NLRP3 receptor need to be transcriptionally/translationally induced, THP-1 cells were first primed with LPS for 4 h, after which solutions with NPs were added. 

We observed the significant increase in IL-1β secretion for P25, DrA, and FG TiO_2_, while for PAA and silver NPs no IL-1β secretion was observed (Figure 8). To confirm that the observed IL-1β secretion was mediated through NLRP3 inflammasome, the experiments were repeated on knock-out THP-1 cells deficient in either Caspase 1 or NLRP3 proteins, which are crucial components of NLRP3 inflammasome. The secretion of IL-1β from caspase-1 and NLRP3 knock-out cells was almost completely diminished compared to wild-type THP-1, confirming the involvement of NLRP3 inflammasome in NPs-triggered IL-1β secretion.

### 2.9. Resolution of Cytokine Secretion

Further, we studied the resolution of cytokine secretion 48 h after the treatment of cells with NPs. THP-1 macrophages were exposed to NPs for 24 h, then NPs were removed and IL-8 and IL-1β secretion was measured after additional 48 h rest. We had not observed significant secretion of cytokines after removal of external NPs for all NP types indicating resolution of the response, except for silver NPs, which still stimulated secretion of IL-8 cytokine, but not IL-1β (Figure 9).

## 3. Discussion

The incidence of allergies and autoimmune diseases has increased over the last few decades [67,68], and alongside various chemicals and environmental pollutants, an increased exposure to NPs is also considered to be a potential risk factor [12,69]. Despite the rapid development of nanomaterials, there are still many unanswered questions regarding NP toxicity and especially NP interactions with the immune system [4,43,44,46,70]. It was shown that some NPs can interact with several components of the immune system and induce different immune responses on a systemic level that are similar to a pseudo-allergic response [71,72]. However, due to the lack of standardization and the complexity of the mechanisms involved in immune-nanotoxicity, there is not enough data available to identify the key physico-chemical properties and/or proteins of the NPs corona that determine the potential nanotoxicity and immunogenicity of various engineered NPs. 

The focus of our study was therefore to determine the correlation between the physicochemical properties of the NPs and the observed biological effects in human THP-1 macrophages. Furthermore, we analyzed whether the protein corona composition modulates the immune response as determined by cytokine secretion, and explored the underlying mechanism of toxicity. We performed a thorough characterization of the physicochemical properties, protein corona composition, and analysis of toxicological endpoints via measurements of viability, ROS induction, and cytokine secretion as a readout method of immune system activation. 

We first analyzed the PCP that defined all subsequent interactions with proteins in the media, as well as with the cells. The NP suspensions exhibited various hydrodynamic diameter distributions in water and culture media, with silver and PAA NPs being the most stable, while most of the engineered NPs exhibiting significant aggregation in the culture media (Figure 1). This can be explained by the high concentration of ions in cell culture media that lowers the zeta potential compared to water, and the binding of the charged proteins (protein corona formation) [59,73,74]. However, in the case of some NPs, the sizes of the aggregates decreased in the media (e.g., TiO_2_ N.B.), suggesting that the protein corona partially stabilized these NPs suspensions. SiO_2_ and DrA TiO_2_ NPs showed most significant aggregation and sedimentation (Z-average > 2000 nm) in water and the culture media (Figure 2), which can be attributed to their low zeta potential. The aggregation of NPs is also evident from the Z-average and PDI measurements (Table 1), as well as on TEM micrographs. In case of the SiO_2_ NPs, where the sedimentation was most pronounced, the DLS method cannot provide accurate measurements, and the obtained results for SiO_2_ in water represent only a small, still stable fraction, while in the cell culture media, almost all SiO_2_ NPs were sedimented due to strong aggregation [53]. 

As the regulatory bodies identified impurities as one of the key parameters of NP toxicity [4,11,40,50,75], we also determined the presence of impurities with ICP-MS and endotoxins using the LAL test. No endotoxins were detected, which is important for the interpretation of the response of THP-1 macrophages, since the presence of LPS itself can stimulate a strong response and thus lead to misleading interpretations. With ICP-MS, we have detected several NPs with various potentially toxic ions, such as arsenic, barium, and nickel; however, as the concentrations of these ions were relatively low (≤1 ng mL^−1^) except for the silver ions in silver NPs, the impurities were probably not the major factor in the observed responses. From our results and the literature [2,75], it then follows that several impurities are present, even in food-grade material such as FG TiO_2_; however, due to the low levels, it is difficult to assess their potential toxicity and immunogenicity.

Apart from the size, surface charge, and surface reactivity, the formation and composition of the protein corona [74,76,77] as the outermost layer of NPs has been proposed to play an important role in the response of immune cells to NP exposure [78,79,80]. The composition of the protein corona depends on the PCP of NPs, as well as on the media in which it is formed [38,53,81,82]. The protein corona is generally divided into hard and soft corona [76,83], and its composition can change when NPs travel through different biological media or cells [84,85]. It is thus crucial to also perform a thorough physicochemical characterization of NPs in physiologically relevant media [59]. 

The protein corona analysis identified significant variability between the corona compositions of different NPs (Figure 3, Appendix A). Importantly, different immune and stress-response proteins were detected, such as complement factor H and heat shock protein HSP 90-α. 

In Table 1, we summarize the main results of the nanotoxicity analysis in terms of the PCP properties, the presence of impurities, detected proteins that are related to the immune system, and toxicological endpoints. In general, except for relatively toxic silver NPs, the lower NP concentrations had no significant effects on the viability of THP-1 macrophages or on ROS induction. No significant decrease in the number of viable cells or increased membrane damage (Figure 4), ROS (Figure 6), or cytokine secretion (Figure 7 and Figure 8) were observed for lower, physiologically achievable NP concentrations (2–10 µg mL^−1^), while higher concentrations of NPs induced different degrees of response, depending on the NP type. Silver NPs were, as expected, the most cytotoxic, followed by TiO_2_ P25, SiO_2_, and TiO_2_ DrA NPs that exhibited moderate cytotoxicity. On the other hand, magnetic PAA, ANATAZ TiO_2_, FG TiO_2_, and N.B. TiO_2_ only slightly affected the viability of macrophages at the highest concentration used (100 µg mL^−1^). Similar results were also obtained for the analysis of oxidative stress (Figure 6). While low NP concentrations (<10 µg mL^−1^) did not induce any changes in ROS, higher concentrations of SiO_2_, P25 TiO_2_, DrA TiO_2_, and N.B. TiO_2_ induced a concentration-dependent increase in ROS. It is important to note that all NPs were internalized, as was determined using TEM imaging (Figure 5), but no NPs were found in the cytosol.

In terms of cytokine secretion, we did not detect IL-6 or TNF-α for any of the selected NPs (Appendix A) following 24 h exposure. Magnetic PAA have altogether not induced any secretion of the tested cytokines (IL-6, TNF-α, IL-8, and IL-1β). However, a dose-dependent secretion of IL-8 (silver NPs), IL-1β (SiO_2_ and N.B. TiO_2_), or both cytokines (P25 TiO_2_, ANATAZ TiO_2_, DrA TiO_2_, and FG TiO_2_) was obtained to various extents (Figure 7 and Figure 8), with a significant increase only at higher NP concentrations (>50 µg mL^−1^) in THP-1 macrophages. In bone marrow-derived mouse macrophage cells, dose-dependent IL-1β secretions were obtained only for P25 TiO_2_ and N.B TiO_2_ (Appendix A), demonstrating variances in response due to interspecies differences. Our results are in agreement with several other studies that also observed IL-1β and/or IL-8 secretion for different TiO_2_ NPs [86,87]. IL-8 is a pro-inflammatory chemokine that plays a key role in the activation of neutrophils, while IL-1β is a known pro-inflammatory chemokine for which an initial induction of pro-IL-1β expression is needed (a priming step). 

Next, we explored the role of the NLRP3 inflammasome in the observed response via the use of caspase-1 and NLRP3 knock-out THP-1 macrophages. IL-1β secretion was completely blunted (Figure 8); thus, we have demonstrated that the observed induction of IL-1β was mediated by NLRP3 inflammasome activation. This is in accordance with studies where NLRP3 activation has been implicated in the inflammatory response to NPs [36,88,89] such as TiO_2_ or SiO_2_ [18,30,31,32,33], and also with studies where it was shown that NPs aggregates can induce NLRP3 activation through several mechanisms, including ROS and lysosome destabilization [64,65,66] leading to IL-1β secretion.

The observed concentration dependence of the THP-1 macrophage response to NPs is important, but it also has to be interpreted carefully with respect to biologically relevant NPs concentrations. Since concentrations of NPs that are 50 µg mL^−1^ or higher are not physiological achievable in vivo, the obtained results for the selected NPs can only be interpreted only as weak immunogenic response in terms of ROS generation and cytokine response. As has already stressed, it is important to also interpret these results in relation to the route of exposure [13,42,90], achievable concentrations, and potential accumulation. For TiO_2_ FG, we can expect the highest exposure in terms of prolonged or continuous uptake with food products, in comparison with cosmetic N.B., DrA TiO_2_, or P25 TiO_2_ NPs where only sporadic and lower-concentration exposures are expected. Moreover, due to the strong barrier of the intestinal epithelium and even the stronger skin barrier, the potential concentrations in the tissue would be much smaller. Indeed, several in vivo studies have demonstrated no immunogenic effect for realistic doses of TiO_2_ or SiO_2_ NPs [46]. On the other hand, the potential accumulation of non-degradable NPs in certain tissues could lead to higher NP concentrations, and as demonstrated by several studies, to chronic inflammation and pathological changes [12,91]. A recent study has suggested that the accumulation of pigment TiO_2_ in the pancreas could be associated with the induction of type II diabetes [12]. 

It is further important to analyze the resolution of cellular responses to NPs. We observed the resolution of cytokine secretion 48 h after the removal of NPs for all NP types (Figure 9) except for silver NPs. Interestingly, the tested formulation of silver NPs still stimulated IL-1β secretion after NP removal. Our results are in agreement with studies where a transient activation of inflammatory pathways and IL-8 secretion was observed for TiO_2_ [29,87], but long-term and in vivo studies are still needed in order to fully understand how the prolonged presence of NPs might trigger inflammatory and immunogenic responses. 

Next, we analyzed possible correlations between PCP properties and toxicological endpoints (Table 2). Noteworthy, there was a very strong correlation between the IL-1β secretion and the size of the NPs (0.59 in water and 0.75 in media). This could be explained by the fact that larger NPs (in term of hydrodynamic diameter) or more aggregated NPs, when internalized, can disrupt several biological processes and/or can lead to the activation of the inflammasome [92,93]. A much smaller correlation of NP size in media (0.44) (but not in water) with IL-8 secretion indicates an additional mechanism of activation for at least certain NPs (silver, ANATAZ TiO_2_, P25 TiO_2_, and SiO_2_).

There were moderate negative correlations between NP size and membrane damage (0.28–0.38), and a moderate positive correlation (0.4) between zeta potential in the medium and observed membrane damage. This suggests that positively charged NPs induce more direct membrane damage, possibly due to their stronger binding to the negatively charged cell membrane, and the proton sponge effect [94]. ROS induction correlated positively with zeta potential (0.59) and PDI in water, indicating that a large size distribution and more positively charged NPs trigger more ROS. Further, ROS induction correlated with IL-1β secretion, which can be explained by the crosstalk between ROS and inflammasome pathways [92]. Still, ROS induction was not observed in the case of DrA TiO_2_ and FG TiO_2_, despite inflammasome activation, but this is in agreement with the literature where it was shown that ROS are not obligatory for inflammasome activation [95]. 

Next, we analyzed all possible correlations between corona protein composition, PCP properties, and toxicological endpoints (Table 3), as most studies stressed the importance of protein corona composition for the toxicological and immunological effects of NPs [13,70,96,97], although it had no effects on NP hemocompatibility [98]. We observed a weak correlation between the NP size in the medium and the total protein count in the corona (Table 3), which is in agreement with studies demonstrating the effects of NP size on corona compositions [70,97]. Interestingly, there was a moderate positive correlation between the total peptide counts (MS counts), ROS induction, and IL-1β secretion, and a strong correlation, with IL-8 indicating that the number of proteins present in the corona positively affects cell stress and the immune response (Table 2). This could partially be due to the increased NP size (the correlation between the size by the number in the media and the total protein count), which also strongly correlated with cytokine secretion, especially IL-1β. Additionally, more bound proteins also increase the probability that the bound proteins are involved in immune processes.

When specific proteins were analyzed in relation to IL-8 and IL-1β secretion, high or moderate correlation coefficients for α1-antiproteinase (A1AT), serum albumin (ALBU), antithrombin-III (ANT3), heat shock protein HSP 90-α (HSP90A), thrombospondin-1 (TSP1), and hemoglobin fetal subunit β (HBBF) were obtained, suggesting a relation. For some, this could be explained by their functions: heat shock proteins are known regulators of inflammatory response, while thrombospondin-1 (TSP1) plays a role in ER stress response via its interaction with activated transcription factor 6α (ATF6). Complement factor H (CFAH) correlated with ROS induction, and to some extent with cytokine secretion (0.57 for IL-8 and 0.36 for IL-1β ) which is in accordance with observed cytokine induction via complement activation [99]. Since A1AT, ALBU, and ANT3 proteins were also detected in the negative control (Figure 3), we cannot make definite conclusions for these proteins. Prothrombin (THRB) correlated strongly with membrane damage. For specific NPs (but not all) the unfolding of fibronectin (FINC), which binds to the surface of NP, has been shown to activate macrophages via the scavenger receptor [100,101], but in our study, we had not observed any specific effects of fibronectin on macrophages. 

Here, we have to stress that due to the limited number of samples and multiple comparisons, the observed correlations (high Spearman coefficients) do not imply causation, but they suggest some relation that has to be explored further in terms of mechanisms. In general, our results suggest that the composition of the protein corona and the presence of specific proteins might be an important factor for physiological responses, but to make definite conclusions, a broader study with a larger number of NPs and human serum is needed. Moreover, the in vitro observations must be supported in vivo [102], since in organisms, NPs are (in addition to other factors) subject to possible degradation, which can affect their toxicity and immunogenicity [103]. 

Altogether, the strong correlation between the NP size in media, and IL-1β and IL-8 secretion suggests that the size of NP aggregates in media is important for governing the responses of macrophages, and this observation is supported by other studies demonstrating the relation between NP size with the toxicity of NPs and inflammasome activation [70,104]. NP-induced secretion of IL-1β was mediated by NLRP3 inflammasome activation, and was mostly accompanied with IL-8 secretion. In the case of silver NPs, Ag ions leakage probably triggered an independent mechanism of IL-8 secretion, while no IL-1β or ROS induction was observed. These observations are important contributions towards uncovering the crucial properties of NPs that govern their potential toxic and immunogenic properties.

## 4. Materials and Methods

### 4.1. Nanoparticles

Experiments were performed with one magnetic and seven types of engineered NPs that were potentially relevant for in vivo exposure. ANATAZ TiO_2_ (Sigma-Aldrich 637254) and P25 TiO_2_ (Sigma-Aldrich 718467) NPs were obtained from Sigma (St Luis, MO, USA), DrA TiO_2_ (Dr. Adorable Inc, 712392053501) from Dr. Adorable Inc (Chicago, Illinois, USA), N.B. TiO_2_ (Naturally Balmy Ltd., NB-2257-1) from Naturally Balmy Ltd. (NB-2257-1, Bournemouth, UK), silver NPs (Biopure 20 nm) were obtained from nanoComposix (San Diego, CA, USA), and engineered SiO_2_ dispersed in 2-propanol were obtained from Nanotesla Institute Ljubljana (Ljubljana, Slovenia). Food-grade (FG) TiO_2_ NPs were obtained from Cake Stuff (Glasgow, UK). All TiO_2_ NP powders were temperature-sterilized, dispersed in ultrapure water, and sonicated.

Biomedical polyacrylic acid (PAA)-coated cobalt ferrite NPs were prepared as described previously [51,105]. Briefly, cobalt ferrite (CoFe_2_O_4_ and Co-ferrite) NPs cores were prepared using the coprecipitation method and stabilized in water. The NP-cores were coated in situ with a 45% (m/m) water solution of polyacrylic acid and sodium salt (PAA) with a molecular weight of 8 kDa (Sigma-Aldrich). The NPs were dialyzed against distilled water and sterilized via filtration using a 0.22 mm pore size syringe filter (Techno Plastic Products TPP, Trasadingen, Switzerland).

### 4.2. Nanoparticle Characterization

Characterization was performed either in water or in RPMI-1640 complete cell culture media (Gibco, Thermo Fisher Scientific, Inc., Waltham, MA, USA) with 10% fetal bovine serum (FBS; Sigma-Aldrich, St Luis, MO, USA). Dynamic light scattering (DLS) was measured using Malvern Zetasizer NanoZS (Malvern Industries, Malvern, UK) with a non-invasive backscatter algorithm. For the measurements of size distribution and of zeta potential, the NP original stocks were first vortexed and then dispersed in either distilled water or cell culture media at the following concentrations: 0.5 *w*/*w*% for PAA, SiO_2_, and silver NPs, and 0.02 *w*/*w*% for all TiO_2_ NPs. The NP suspensions were re-suspended several times with a pipette. The Z-average size from intensity distribution, polydispersity index (PDI), and hydrodynamic diameter (peak) based on number distribution of particles are reported. The DLS measurements were performed using 20 consecutive runs on individual samples at 30 s each. The zeta potential was also measured on the Zetasizer NanoZS, with disposable folded capillary cells and M3-PALS measurement technology. The measurement was conducted after 5 min of stirring the colloidal suspension in the sample cell. A refractive index of 1.10 was used. NP suspensions were tested for the presence of endotoxins (PYROGENT™ Plus Gel Clot LAL Assays, Lonza Group Ltd., Basel, Switzerland). The endotoxin concentration was lower than 0.5 EU mL^−1^ for all NP suspensions tested.

### 4.3. ICP-MS Analysis of Ions Leaching from NPs

The NPs (30 µg mL^−1^) were aged for 24 h at 36.5 °C in physiological solution (0.9% NaCl), and then separated from the suspensions using sequential filtration. The filtrate was ultrafiltrated using 3 kD (~1–2 nm pore size) Amicon Ultra-4 Centrifugal Filter Units (Merck Millipore, Burlington, MA, USA) and centrifuged at 800× *g* rpm for 30 min to separate the particles from the dissolved fraction. The concentrations of the ions in the filtered solutions were determined via inductively coupled plasma MS (ICP-MS) (Agilent 7900, Agilent Technologies, Tokyo, Japan), as described elsewhere [2].

### 4.4. NP Corona Preparation

RPMI-1640 complete cell culture medium with 10% FBS and 2 mM glutamine (Sigma-Aldrich, St. Luis, MO, USA) was used to prepare the protein corona of the NPs. LoBind micro-centrifuge tubes (Eppendorf, Hamburg, Germany) were used for all steps of the protocol. Initial NP dispersions (1 μg μL^−1^) were prepared in distilled water, vortexed, and incubated for 5 min at room temperature. Afterwards, 100 μL of NP dispersion (containing 100 μg of NPs) was added to 900 μg of complete cell culture media, vortexed, and incubated for 1 h at 37 °C. The FBS-NPs mixtures were transferred to a new micro-centrifuge tube and centrifuged at 15,000× *g* for 20 min at 4 °C. The supernatants were removed, and the pellets were dispersed in 1 mL of cold PBS without CaCl_2_ and MgCl_2_. This procedure was repeated three times. The proteins that remained adhered to the NPs were considered to present the hard protein corona of the NPs [14]. The control samples were prepared by following the same steps with an equal volume of distilled water added to the complete cell culture media instead of NP suspensions.

### 4.5. Sodium Dodecyl Sulphate Polyacrylamide Gel Electrophoresis (SDS-PAGE) and Mass Spectrometry (MS)

Following the last centrifugation step during protein corona preparation (see Section 4.4), the supernatants were removed and NPs were suspended in 100 μL of non-reducing sodium dodecyl sulphate (SDS) sample buffer (10% (*m*/*v*) SDS, 25% (*v*/*v*) glycerol, 0.5% (*m*/*v*) Bromophenol Blue, and 300 mM TRIS-HCl, pH 8.8). The suspension was vortexed and heated for 5 min at 95 °C. NP–protein complexes were thus shattered, and the NPs were removed via centrifugation at 15,000× *g* for 20 min. Aliquots of 20 μL were loaded onto 10% (*m*/*v*) SDS–polyacrylamide gel in SDS running buffer (10 g SDS, 30.3 g TRIS, and 144 g glycine in 10 L of dH_2_O) and resolved at 195 V. After electrophoresis, the gels were stained with AgNO_3_ and developed using Na_2_CO_3_.

Protein identification was performed using mass spectrometry (MS) analysis, as described previously [53]. Proteins were in-gel trypsinized, and the resulting peptides were extracted. The extracts were purified on in-house packed C18 StageTips (Empore extraction disks C18, 3 M, St. Paul, Minneapolis, USA) and analyzed using a liquid chromatography (LC)–electrospray ionization (ESI) ion trap-MS/MS (MSD Trap XCT Plus, Agilent Technologies, Waldbronn, Germany) as previously described [106]. The LC-ESI-MS/MS spectral data were searched against the SwissProt database using an in-house Mascot search engine (version 2) with the following parameters: two missed cleavages were allowed, peptide and fragment mass tolerances of 1.2 and 0.6 Da, respectively, were used, carboxyamidomethylcysteine (C) was used as a fixed modification, and oxidized methionine was used as a variable. The results were validated using Scaffold 2 software (Proteome Software, Portland, Oregon, USA). Spectral counts matching to a protein are an indicator of its amount in a given sample [107], and exclusive spectral counts were used as a measure of protein abundance, and to calculate relative protein abundance [53]. The sum of the relative protein abundances of all proteins over a given sample is 100%. Protein corona analysis was performed in three independent repeats. Only proteins that were detected in at least two repeats were listed and included in the statistical analysis. 

### 4.6. Cell Culture

The THP-1 cell line was acquired from ECACC (Salisbury, UK). Cells were cultured in RPMI-1640 cell culture media with 10% (*v*/*v*) FBS and 2 mM glutamine. The differentiation of THP-1 cells to macrophage-like cells was performed via their exposure to 162 nM phorbol 12-miristrat 13-acetate (PMA; Sigma-Aldrich, St. Luis, MO, USA) for 72 h. When differentiated, the cells became attached to the bottom of the culturing vessels. Following differentiation, the cells were washed to remove the differentiation medium, and left in RPMI + 10% (*v*/*v*) FBS for 24 h. They were then incubated with NPs for different experiments. 

THP-1 deficient in NLRP3 and caspase-1 were a kind gift of Veit Hornung (LMU Munich, Germany). Those cells were prepared using CRISPR/cas9 technology, as described in [108]. Caspase-1 mediates the non-canonical activation of the NLRP3 inflammasome in human myeloid cells. Immortalized bone marrow-derived mouse macrophages from C57BL/6 mice (iBMDMs) were a gift of Kate A. Fitzgerald (University of Massachusetts Medical School, USA) [89]. iBMDMs cell were cultured in DMEM supplemented with 10% (*v*/*v*) FBS.

### 4.7. Viability

The differentiated cells were exposed to increasing concentrations of NPs for 24 h. The NPs original stocks were first vortexed and dispersed. After the NPs were added to the cell media, the suspensions were re-suspended several times with a pipette. After 24 h incubation, cell viability was determined via differential staining with 2 μg mL^−1^ Hoechst 33,342 (Life Technologies, CA, USA) to determine the total cell number, and with 0.15 mM propidium iodide (PI; Sigma-Aldrich) for 5 min to stain dead cells. The fluorescent images obtained were analyzed with CellCounter software [47]. The cell viability percentage (%Viability) was determined as the ratio between the number of viable cells in each sample (NS = Hoechst positive-PI positive) and the number of all cells in the non-treated control (N0): %Viability = 100 × NS/N0. The number of dead cells (%dead) was calculated as %Dead = 100 × Number PI positive/Number Hoechst positive) [51,52]. 

### 4.8. ROS Assay

ROS levels were determined with the 5-(and-6)-chloromethyl-2′,7′-dichlorodihydrofluorescein diacetate assay (CM-H_2_DCFH-DA; Molecular Probes, Invitrogen). After a 24 h incubation with increasing concentrations of NPs, the cells were washed and incubated with 10 μM CM-H_2_DCF-DA in DMEM FluoroBrite (Gibco, Thermo Fisher Scientific) at 37 °C for 45 min. We used 0.1 mM H_2_O_2_ (20 min incubation) as a positive control. The fluorescence was measured using a Tecan Infinite 200 spectrofluorometer (Tecan, Männedorf, Switzerland). The fluorescence intensity of CM-H_2_DCFH-DA was normalized to the relative number of cells, as determined by Hoechst 33342. The results are presented as the percentage and standard error of normalized fluorescence intensity compared to the negative control sample for three independent experiments in three replicates.

### 4.9. Transmission Electron Microscopy

For the observation of NP internalization, cells were grown in 40 mm Petri dishes (TPP AG, Switzerland) and incubated with NPs for 24 h in full cell culture medium. Following incubation, cells were washed to remove loose NPs, and fixed with a mixture of 4% (*m*/*v*) paraformaldehyde and 2% (*v*/*v*) glutaraldehyde in 0.1 M cacodylate buffer, pH 7.4, for 2 h at 4 °C. Post-fixation was conducted in 1% osmium tetroxide in 0.1 M cacodylate buffer for 1 h, followed by dehydration in graded ethanol and embedding in Epon 812 resin (Serva Electrophoresis, Heidelberg, Germany). Ultra-thin sections were contrasted with uranyl acetate and lead citrate, and examined with a Philips CM100 TEM. 

For the additional characterization of NPs, nanoparticle suspensions were re-suspended in 100 µL of distilled water with the following final concentration: 200 µg/mL. Then, 5 µL of this suspension was added onto a copper grid covered with a formwar foil. After drying at room temperature, the samples were observed with a Philips transmission electron microscope, CM100 TEM (Appendix A).

### 4.10. ELISA

For the detection of the cytokines IL-8, IL-6, TNF-α, and IL-1β (for IL-6 and TNF-α, refer to the Appendix A) we used an immuno-enzyme test (ELISA; Affymetrix eBioscience Inc., Atlanta, GA, USA) following the manufacturers’ instructions. For IL-1β, the detection cells were first primed with 50 ng mL^−1^ LPS for 5 h, after which, NPs were added for 24 h. Mouse iBMDMs were primed with 100 ng mL^−1^ LPS for 6 h and afterwards they were stimulated with NP solution. A concentration of 100 ng mL^−1^ LPS and 200 µg mL^−1^ SiO_2_ NPs (NLRP3 Inflammasome Inducer, Invivogen) were used as the positive controls. To observe the resolution of immune activation, the NPs were washed following 24 h incubation, and cytokine concentration was measured after an additional 48 h.

### 4.11. Statistics

The results are presented as mean and standard error of the mean. A one-way analysis of variance (ANOVA) with Dunnett’s correction for multiple comparisons was performed to test for statistical differences between the control and treated samples. Statistical analyses were performed using GraphPad Prism v6 (GraphPad Prism Software, La Jolla, CA). Statistical significance is displayed as follows: ns (*p* > 0.05); * *p* ≤ 0.05; ** *p* ≤ 0.01; *** *p* ≤ 0.001, **** *p* ≤ 0.0001. Spearman’s coefficient was used to measure the correlation between variables. Statistical significance was determined using a permutation test. The *p*-values were corrected for multiple testing using Bonferroni correction.

## 5. Conclusions

Understanding the relation between NP physicochemical properties and their toxicity/immunogenicity is important for the identification of potential harmful engineered NPs. In the present paper, we performed an in vitro nanotoxicity study on THP-1 macrophages of eight NPs that are commonly used in pigments, food, and cosmetic products, and analyzed the correlation between PCP properties, protein corona composition, cytotoxicity, and immunogenicity. 

Dose dependent cytotoxicity was exhibited by silver and SiO_2_ NPs, while significant ROS induction was observed for higher concentrations (>50 µg mL^−1^) of P25, DrA, N.B. TiO_2_, and SiO_2_. Several TiO_2_ and SiO_2_ NPs induced the transient secretion of pro-inflammatory IL-8 (P25, DrA, ANATAZ, FG TiO_2_, and SiO_2_) and/or IL-1β (FG and N.B. TiO_2_) cytokines, while for silver NPs, the cytokine secretion response persisted even after the removal of NPs. Lower concentrations of all of the tested NPs exhibited relatively low cytotoxicities and weak stress/cytokines responses; however, non-degradable NPs can be accumulated at different sites and organs, triggering inflammation, especially in the case of the food-grade NPs.

We showed that cell responses strongly depend on NP properties. We identified NLRP3 inflammasome activation as being the main factor that induced IL-1β secretion. Correlation analysis suggested that the hydrodynamic size of NPs and the amount of protein present in the protein corona, as well as the presence of certain proteins, can significantly affect cell stress and the immune response. More generally, this is important since it demonstrates that NP (hydrodynamic) size in physiological media could be one of the fundamental parameters that has to be considered when evaluating the potential immunogenicity of specific NPs. Furthermore, correlations analysis implies that the presence of certain proteins in the protein corona, such as heat shock proteins and proteins that are involved in ER stress and the complement system could potentially induce immune responses; however, this should be investigated further in order to specifically test the role of each protein. 

In conclusion, our study is important for understanding the mechanisms of toxicity and immunogenicity for engineered NPs to which we are chronically exposed, especially since TiO_2_ NPs are usually considered to be non-toxic. 

Many studies have shown that chronic stress and pro-inflammatory responses are associated with different pathologies such as cancer and neurodegenerative diseases. Even in the absence of acute cytotoxicity at the cellular level, elevated cytokines can trigger an immune response and lead to long-term harmful immunotoxicity by impairing immune cell function. 

## Figures and Tables

**Figure 1 ijms-23-06197-f001:**
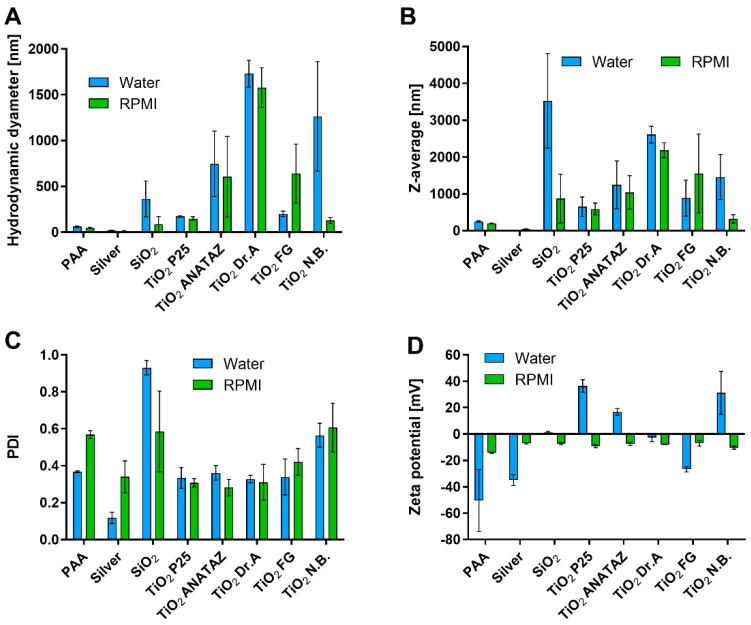
Analysis of physico-chemical properties of NPs. (**A**) Hydrodynamic diameter based on the particle number distribution; (**B**) Z-average size based on the intensity distribution and (**C**) polydispersity index (PDI) of the number distribution measured with DLS; (**D**) Zeta potential in water and RPMI media with 10% FBS (*m*/*v*). Mean values with standard error of the mean from three independent measurements are shown.

**Figure 2 ijms-23-06197-f002:**
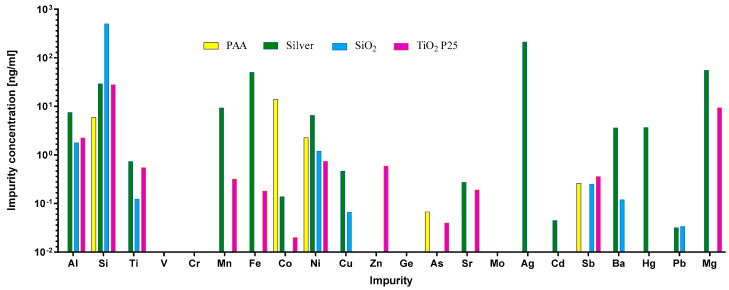
ICP-MS analysis of the inorganic impurities (Al, Si, Ti, V, Cr, Mn, Fe, Co, Ni, Cu, Zn, Ge, As, Sr, Mo, Ag, Cd, Sb, Ba, Hg, and Pb) detected in NP suspensions after 24 h of leaching in 0.9% (*m*/*v*) NaCl. The measured concentrations of the elements are given in ng mL^−1^.

**Figure 3 ijms-23-06197-f003:**
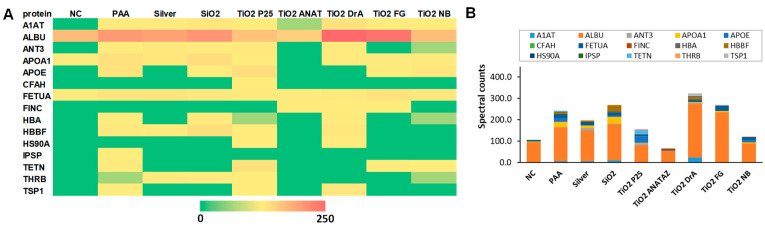
Composition of protein corona after 1 h of NP incubation in RPMI-1640 media with 10% (*m*/*v*) FBS. Proteins were separated from NPs, analyzed using SDS-PAGE, and identified via MS. (**A**) Heatmap of protein abundance (spectral counts obtained from MS) and (**B**) protein abundance. the average values of three independent repeats are presented (see also Appendix A). Full names of the proteins are provided in Appendix A.

**Figure 4 ijms-23-06197-f004:**
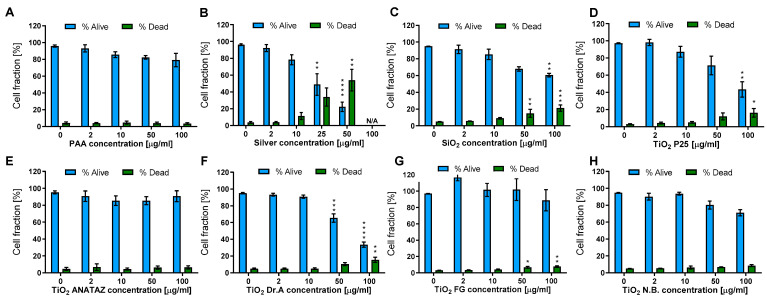
THP-1 viability and membrane damage following 24 h incubation with (**A**) polyacrylic acid (PAA)-coated NPs, (**B**) Ag NPs, (**C**) engineered SiO_2_ NPs, (**D**) TiO_2_ P25, (**E**) TiO_2_ ANATAZ NPs, (**F**) TiO_2_ DrA NPs, (**G**) TiO_2_ food-grade (FG), and (**H**) TiO_2_ N.B. NPs. The viability was determined with Hoechst/PI differential staining. Please note that lower concentrations were used for silver NPs. Mean and standard error are shown for three independent experiments. Statistical significance is displayed as follows: ns (*p* > 0.05); * *p* ≤ 0.05; ** *p* ≤ 0.01; *** *p* ≤ 0.001, **** *p*≤ 0.0001.

**Figure 5 ijms-23-06197-f005:**
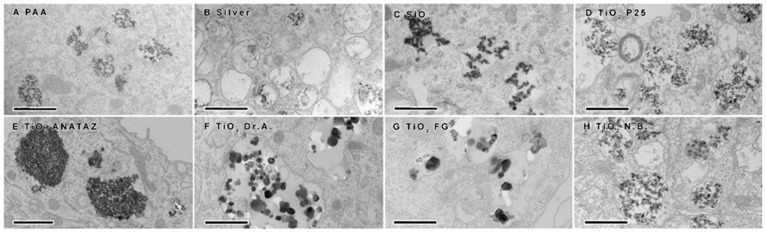
Transmission electron microscopy analysis of differentiated THP-1 cells following 24 h incubation with 50 µg mL^−1^ (25 µg mL^−1^ for silver NPs) of (**A**) polyacrylic acid (PAA)-coated magnetic NPs, (**B**) silver NPs, (**C**) engineered SiO_2_ NPs, (**D**) TiO_2_ P25, (**E**) TiO_2_ ANATAZ, (**F**) TiO_2_ DrA, (**G**) TiO_2_ food-grade (FG), and (**H**) TiO_2_ N.B. NPs. Scale bars correspond to 600 nm.

**Figure 6 ijms-23-06197-f006:**
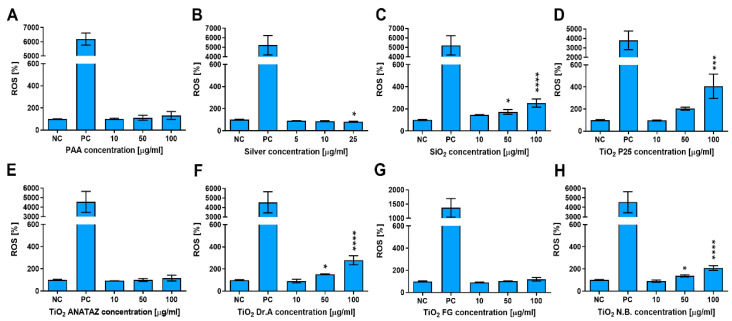
ROS induction in THP-1 macrophages following 24 h incubation with increasing concentrations of (**A**) polyacrylic acid (PAA)-coated NPs, (**B**) silver NPs, (**C**) engineered SiO_2_ NPs, (**D**) TiO_2_ P25, (**E**) TiO_2_ ANATAZ NPs, (**F**) TiO_2_ DrA NPs, (**G**) TiO_2_ food-grade (FG), and (**H**) TiO_2_ N.B. NPs, as determined with CM-H_2_DCFH-DA assay. For the positive control (PC), cells were incubated with 1 mM H_2_O_2_ for 20 min. Please note that lower concentrations were used for silver NPs. Mean and standard error of three independent experiments are shown. Statistical significance is displayed as follows: ns (*p* > 0.05); * *p* ≤ 0.05; *** *p* ≤ 0.001, **** *p*≤ 0.0001.

**Figure 7 ijms-23-06197-f007:**
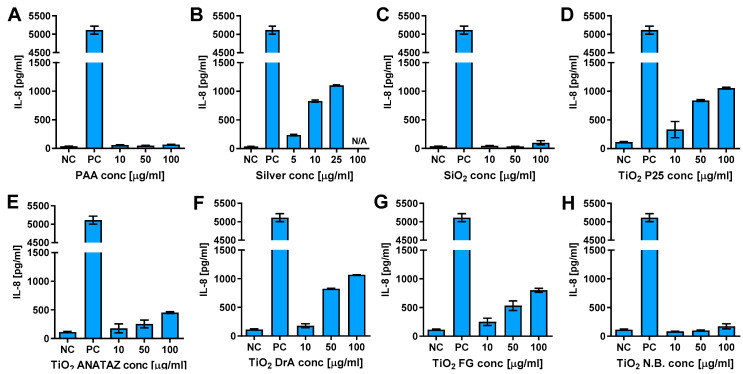
IL-8 secretion from THP-1 macrophages following 24 h incubation with increasing concentrations of (**A**) polyacrylic acid (PAA)-coated NPs, (**B**) silver NPs, (**C**) engineered SiO_2_ NPs, (**D**) TiO_2_ P25, (**E**) TiO_2_ ANATAZ NPs, (**F**) TiO_2_ DrA NPs, (**G**) TiO_2_ food-grade (FG), and (**H**) TiO_2_ N.B. NPs, as determined using ELISA. Cell exposed to 100 ng mL^−1^ LPS for 24 h were used as a positive control. Please note, that lower concentrations were used for silver NPs due to extensive cells death. Representative experiment of the three experiments is shown. Mean and standard error of three replicates of the same biological repeat are shown.

**Figure 8 ijms-23-06197-f008:**
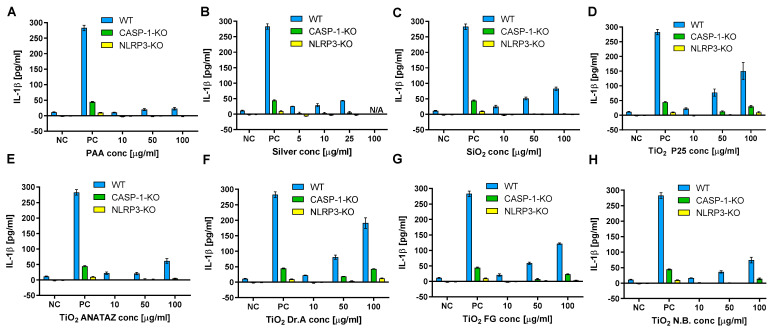
IL-1β secretion in THP-1 macrophages following 18 h incubation with increasing concentration of (**A**) PAA coated NPs, (**B**) silver NPs, (**C**) engineered SiO_2_ NPs (**D**) TiO_2_ P25 NPs, (**E**) TiO_2_ ANATAZ NPs, (**F**) TiO_2_ DrA NPs, (**G**) TiO_2_ food-grade (FG) NPs and (**H**) TiO_2_ N.B. NPs as determined with ELISA. THP-1 macrophages were primed with LPS for 4 h, after which exposure to NPs was performed. A concentration of 200 µg mL^−1^ crystalline SiO_2_ NPs was used as a positive control. Please note, that lower concentrations were used for silver NPs due to excessive cell death. Representative experiment of the two experiments is shown. Mean and standard error of three replicates of the same biological repeat are shown.

**Figure 9 ijms-23-06197-f009:**
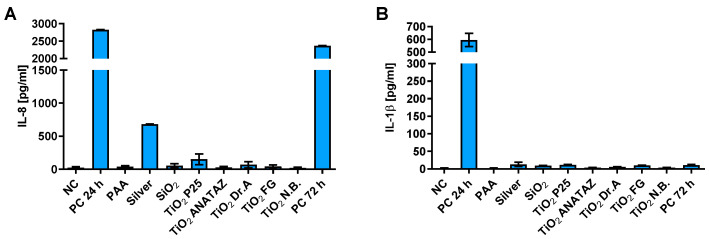
Resolution of (**A**) IL-8 and (**B**) IL-1β cytokine secretion 48 h after 24 h incubation of THP-1 macrophages with 50 µg mL^−1^ of NPs (25 µg mL^−1^ for silver) as determined with ELISA. A concentration of 100 ng mL^−1^ LPS for IL-8 or 200 µg mL^−1^ crystalline SiO_2_ NPs for IL-1β were used as a positive control with 24 h or 72 h incubation, for PC 24 h and PC 72 h, respectively. Please note the different scale on the y-axes for panels A and B.

**Table 1 ijms-23-06197-t001:** Overview of the NPs PCP, corona proteins (involved in the immune response) and the cell responses to NP exposure.

Parameter\NP Type		PAA	Silver	SiO_2_	TiO_2_ P25	TiO_2_ ANAT	TiO_2_ DrA	TiO_2_ FG	TiO_2_ N.B.
**PCP**	**Zeta potential_RPMI_ [mV]**	−14 ± 0.2	−7.0 ± 0.3	−7.3 ± 0.6	−9.3 ± 0.9	−7.4 ± 1.0	−8.1 ± 0.1	−8.2 ± 1.7	−10.3 ± 1.03
	**Z-average_RPMI_ [nm]**	199 ± 6	48 ± 2.6	871 ± 547 ^a^	589 ± 131	1040± 375	2182 ± 166	1552 ± 884	321 ± 95
**Impurities**	**Endotoxin**	NO	NO	NO	NO	NO	NO	NO	NO
	**Detected impurities**	Mn, Co, Ti, Zn, Mg	Ag, Fe, Ba, Mn	Al, Ti, Fe, Zn, Pb	Mg, Al, Si, Ti, Mn, Fe, Co, Zn, Sr, Sb, As	Zn, Mg	Zn, Mg, Fe, Mn, V	Mg, Al, Ti, Mn, Fe, Ni, Zn, Ge, As, Sb, Ba, Pb	Zn, Mg, Fe,
**Protein Corona**	**HSP90A**	NO	NO	NO	YES	NO	YES	NO	NO
	**FINC**	NO	NO	NO	NO	YES	YES	YES	NO
	**CFAH**	NO	NO	NO	YES	NO	NO	NO	YES
**Toxicity mechanisms**	**IC90 [µg mL^−1^]**	13	2.5	2.9	12	4.2	9.1	103	15
	**Membrane damage**	NO	YES	YES	YES	NO	YES	NO	NO
	**Internalization**	YES	YES	YES	YES	YES	YES	YES	YES
	**ROS**	o	o	**+**	**+**	o	**+**	o	**+**
**Cytokine response**	**TNF-α**	o	o	o	o	o	o	o	o
	**IL-6**	o	o	o	o	o	o	o	o
	**IL-8**	o	+	o	+	+	+	+	o
	**IL-1β**	o	o	+	+	+	+	+	o

^a^ SiO_2_ NPs formed large aggregates >μm in diameter at pH 7, which quickly sedimented so that the DLS measurements only represented a smaller fraction of these NPs.

**Table 2 ijms-23-06197-t002:** Correlations (Spearman coefficient) between NP PCP properties: Size (hydrodynamic diameter D_hyd_—peak by number distribution) in distilled water (H_2_O) and RPMI media (RPMI) Z-average size (Z-av), zeta potential (Zeta), PDI, and toxicological endpoints: IC90, membrane damage (Memb. dam.), ROS, IL 8, and IL-1β.

	D_hyd_ H_2_O	D_hyd_ RPMI	Z-aver H_2_O	Z-aver RPMI	Zeta H_2_O	Zeta RPMI	PDI H_2_O	PDI RPMI	IC90	Mem dam.	ROS	IL-8	IL-1β
**D_hyd_ H_2_O**	1	0.72	0.55	0.58	0.44	0.11	0	−0.3	−0.2	−0.39	0.12	0.01	0.58
**D_hyd_** **RPMI**	0.72	1	0.4	0.93	0.06	0.31	−0.08	−0.2	0.17	−0.28	0.05	0.43	0.75
**Z-av H_2_O**	0.55	0.4	1	0.56	0.31	0.37	0.77	0.27	−0.19	−0.34	0.44	−0.06	0.5
**Z-av** **RPMI**	0.58	0.93	0.56	1	0.07	0.45	0.19	0.01	0.35	−0.38	0.17	0.34	0.71
**Zeta H_2_O**	0.44	0.06	0.31	0.07	1	0.23	0.26	−0.31	−0.26	−0.22	0.6	0.08	0.42
**Zeta** **RPMI**	0.11	0.31	0.37	0.45	0.23	1	0.17	−0.12	0.17	0.43	0	0.37	0.26
**PDI H_2_O**	0	−0.08	0.77	0.19	0.26	0.17	1	0.67	−0.18	−0.4	0.52	−0.27	0.16
**PDI med**	−0.3	−0.2	0.27	0.01	−0.31	−0.12	0.67	1	−0.08	−0.38	−0.15	−0.56	−0.37
**IC90**	−0.2	0.17	−0.19	0.35	−0.26	0.17	−0.18	−0.08	1	−0.22	−0.07	0.03	−0.05
**Mem.dam**	−0.39	−0.28	−0.34	−0.38	−0.22	0.43	−0.4	−0.38	−0.22	1	−0.3	0.45	−0.2
**ROS**	0.12	0.05	0.44	0.17	0.6	0	0.52	−0.15	−0.07	−0.3	1	0.36	0.66
**IL-8**	0.01	0.43	−0.06	0.34	0.08	0.37	−0.27	−0.56	0.03	0.45	0.36	1	0.69
**IL-1b**	0.58	0.75	0.5	0.71	0.42	0.26	0.16	−0.37	−0.05	−0.2	0.66	0.69	1

**Table 3 ijms-23-06197-t003:** Analysis of correlations (Spearman coefficient) between the protein corona composition and PCP: total protein abundance (PROT_ab_), D_hyd_ (diameter—peak by number distribution) in distilled water (H_2_O) and RPMI media (RPMI), Z-average size (Z-aver), zeta potential (Zeta), PDI, and toxicological endpoints: IC90, membrane damage (Mem dam), ROS, IL-8, and IL-1β.

	D_hyd_ H_2_O	D_hyd_ RPMI	Z-aver H_2_O	Z-aver RPMI	PDI H_2_O	PDI RPMI	Zeta H_2_O	Zeta RPMI	IC90	Mem dam	ROS	IL-8	IL-1β
**PROT_ab_**	−0.09	0.27	0.14	0.29	0.01	−0.31	−0.22	−0.02	0.21	0.11	0.53	0.71	0.6
**A1AT**	0.59	0.75	0.47	0.6	−0.06	−0.32	−0.11	0.17	−0.21	0.08	0.16	0.57	0.74
**ALBU**	−0.05	0.4	0.03	0.41	−0.14	−0.43	−0.17	0.09	0.4	0.1	0.48	0.79	0.65
**ANT3**	0.21	0.39	−0.03	0.17	−0.34	−0.66	0.15	0.18	−0.33	0.48	0.31	0.9	0.68
**APOA1**	−0.3	−0.4	0.47	−0.22	0.68	0.57	−0.33	−0.14	−0.23	−0.02	0.19	−0.32	−0.21
**APOE**	−0.16	−0.49	−0.14	−0.5	0.09	0.27	−0.24	−0.89	−0.06	−0.47	0.03	−0.63	−0.43
**CFAH**	−0.26	−0.19	−0.32	−0.24	−0.1	−0.43	0.51	−0.08	−0.09	0.14	0.65	0.57	0.36
**FETUA**	−0.61	−0.44	−0.48	−0.38	−0.36	−0.23	−0.56	−0.2	0.56	0.31	−0.12	0.06	−0.39
**FINC**	0.13	0.32	0.07	0.41	0.19	0.48	0.17	0.34	0.1	−0.31	−0.29	−0.22	−0.04
**HBA**	0.13	0.24	0.72	0.01	0.87	0.46	0.06	0.26	−0.21	−0.01	0.45	−0.13	0.07
**HBBF**	0.51	0.59	0.75	0.57	0.38	0.01	−0.04	0.16	−0.3	−0.08	0.37	0.38	0.71
**HS90A**	0.03	0.15	−0.15	0.04	−0.15	−0.55	0.51	−0.02	−0.14	0.09	0.68	0.75	0.65
**IPSP**	−0.35	−0.3	−0.43	−0.43	−0.51	−0.26	−0.4	0.3	−0.21	0.93	−0.61	0.19	−0.44
**TETN**	−0.37	−0.25	−0.45	−0.25	−0.19	0.27	−0.59	−0.8	0.38	−0.42	−0.2	−0.38	−0.4
**THRB**	−0.46	−0.51	−0.45	−0.63	−0.41	−0.37	−0.15	0.22	−0.33	0.95	−0.28	0.29	−0.35
**TSP1**	0.67	0.82	0.32	0.64	−0.17	−0.48	0.17	0.11	−0.16	−0.06	0.32	0.68	0.89

## Data Availability

The data presented in this study are available in the article or in the Appendix A.

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
