# Peer review of "The Relevance of Physico-Chemical Properties and Protein Corona for Evaluation of Nanoparticles Immunotoxicity—In Vitro Correlation Analysis on THP-1 Macrophages"

_ijms, 2022, doi:10.3390/ijms23116197_

Round 1
Reviewer 1 Report
In this work, the authors evaluated the role played by the physicochemical properties of several commonly used nanoparticles [P25 industrial grade TiO2, food grade TiO2, cosmetic grade TiO2, SiO2, silver and 1 polyacrylic acid (PAA) coated magnetic NPs], specifically the formation and composition of the protein corona, in the immune cells response, namely in differentiated THP-1 macrophages. With this aim, the correlation between the physicochemical properties (hydrodynamic diameter, PDI, Z-average and zeta potential) and the protein corona composition of the different NPs was assessed, based on cyto- and immunotoxicity measurement endpoints such as cell viability, reactive oxygen species (ROS) production, and cytokine secretion. It is well known that common and/or standard toxicity tests might not be fully adequate for evaluating nanomaterials, since their unique features may be also responsible for unexpected interactions with assay components or detection systems. The lack of specific standardized protocols for nanotoxicology, the huge diversity of both NPs and testing experimental conditions, and the scarce number of studies in the literature on the potential immunotoxic effect of NPs, makes it imperative to realistically know which physicochemical properties are the main players in the immune response after exposure, and the role in toxicity of nano-bio interactions on in vivo relevant conditions.
In other hand, the authors present a well-structured and well written manuscript. The study is technically sound and correctly designed. Thanks to this, the authors demonstrated that all tested TiO2 types and SiO2, except silver NPs, showed moderate toxicity and transient inflammatory response, observed as an increase in ROS, IL-8 and/or IL-1β cytokine secretion. The correlation analysis suggested that hydrodynamic size of NPs, and the amount and association of specific NP corona proteins, can significantly affect cellular stress and immune response in terms of cytokine secretion induction. Also, authors identified NLRP3 inflammasome activation as a main factor that induced IL-1β secretion. Broadly, results showed that the NPs size in physiological media could be one of the key parameters to regard when assessing potential immunogenicity of specific NPs.
The paper is of potential interest to the community. However, some minor issues should be addressed by the authors before publication:
Page 2 - Line 78: Please, insert brackets as follows, so that sentence is expressed correctly, “…products [P25 industrial grade TiO2, food grade TiO2, cosmetic grade TiO2, SiO2, silver and 1 polyacrylic acid (PAA) coated magnetic NPs].”
Page 2 - In the sentence of lines 86-88: “Importantly, the cytokine secretion strongly correlated with the size of the NPs in the culture media, while specific proteins in the protein corona only had moderate effects.” Reformulate the sentence to clarify the meaning of “only moderate effect of specific protein corona composition” in the cytokine secretion.
Page 2-3 – Lines 97-100: Authors should introduce references for the applications of the NPs listed below, ”c) Silver NPs as NPs widely used in various materials (citation), d) P25 (citation) and e) ANATAZ TiO2 a reference TiO2 material (citation), f) Dr. Adorable (DrA) TiO2 (citation) and g) Naturally Balmy (N.B.) TiO2 NPs used in cosmetics (citation), and h) Food-grade (FG) TiO2 that is used in food industry products (citation).”
I suggest some articles in case they are useful:
Yusuf M. (2018). Silver Nanoparticles: Synthesis and Applications. Handbook of Ecomaterials, 2343–2356. https://doi.org/10.1007/978-3-319-68255-6_16
Zhang X. F., Liu Z. G., Shen W., & Gurunathan S. (2016). Silver Nanoparticles: Synthesis, Characterization, Properties, Applications, and Therapeutic Approaches. International journal of molecular sciences, 17(9), 1534. https://doi.org/10.3390/ijms17091534
Zivic F., Grujovic N., Mitrovic S., Ahad I.U., Brabazon D. (2018). Characteristics and Applications of Silver Nanoparticles. In: , et al. Commercialization of Nanotechnologies–A Case Study Approach. Springer, Cham. https://doi.org/10.1007/978-3-319-56979-6_10
Jafari S., Mahyad B., Hashemzadeh H., Janfaza S., Gholikhani T., Tayebi L. (2020). Biomedical Applications of TiO2 Nanostructures: Recent Advances. Int J Nanomedicine, 15, 3447-3470.
https://doi.org/10.2147/IJN.S249441
Nicosia, A., Vento, F., Di Mari, G.M., D’Urso, L., Mineo, P.G. (2021). TiO2-Based Nanocomposites Thin Film Having Boosted Photocatalytic Activity for Xenobiotics Water Pollution Remediation. Nanomaterials, 11, 400. https://doi.org/10.3390/nano11020400
Jiang X., Manawan M., Feng T., Qian R., Zhao T., Zhou G., Kong F., Wang Q., Dai S., Pan J. H. (2018). Anatase and rutile in evonik aeroxide P25: Heterojunctioned or individual nanoparticles? Catalysis Today, 300, 12–17. http://dx.doi.org/10.1016/j.cattod.2017.06.010
Adawiyah J. Haider, Zainab N. Jameel, Imad H.M. Al-Hussaini. (2019). Titanium Dioxide Applications. Energy Procedia, 157, 17-29. https://doi.org/10.1016/j.egypro.2018.11.159.
Dréno B., Alexis A., Chuberre B., Marinovich M. (2019). Safety of titanium dioxide nanoparticles in cosmetics. Journal of the European Academy of Dermatology and Venereology (JEADV), 33 (7), 34–46. https://doi.org/10.1111/jdv.15943
Baranowska-Wójcik E. (2021). Factors Conditioning the Potential Effects TiO2 NPs Exposure on Human Microbiota: a Mini-Review. Biological trace element research, 199(12), 4458–4465. https://doi.org/10.1007/s12011-021-02578-5
Page 4 – Lines 140-142: In figure 2, authors should indicate the measured parameter and units of the Y-axis and include this information in the figure legend.
Page 6 – Lines 208-210: In figure 5, the alphabetical index of NPs is not correct. Silver NPs have been omitted, thereby, adjust the entire figure caption accordingly. Differences on incubation conditions for these NPs (25µg.ml-1 for 24h) must be included.
Page 7 – Lines 220-223: Authors should specify and add in the figure legend of Figure 6 which substance was used as positive control in the quantification of % ROS, and its concentration and incubation time.
Page 8 – Lines 232-236: In figure 7, add the meaning of N/A in the figure caption. On the other hand, data for cytokine secretion would be more robust if accompanied by statistical analysis.
Page 10 – Lines 310-311: The following sentence, “With ICP MS we have detected several NPs various potentially toxic ions, such as arsenic, barium and nickel.” needs to be revised and rewritten. It seems that some preposition is missing to make it sense.
Page 10 and 11 – from Line 317: Dimension of Table 1 or the font size need to be adjusted, in order that all items can be fully read.
Page 12 – Lines 373-375: The position of the three references (cites 13, 42, 86) does not seem to be quite appropriate. I suggest moving them just before the comma.
Page 12 – Lines 380-381: Check the punctuation of the following sentence “Indeed. several in vivo studies demonstrated no immunogenic effect for realistic doses of TiO2 or SiO2 NPs46.”
Page 14 – Line 479: It would be interesting to specify and include in the methodology the dispersion protocol used by the authors, since its importance for results reproducibility is well known in nanotoxicology, and since the NPs size is one of the key properties in which the conclusions of the study is based on.
Page 17 – Conclusions: In general, this section is quite long. The authors should avoid repeating the description of results in some paragraphs, and only expose the conclusions drawn from the study outcomes.
Author Response
We thank the reviewer for positive comments and suggestions.
Page 2 - Line 78: Please, insert brackets as follows, so that sentence is expressed correctly, “…products [P25 industrial grade TiO2, food grade TiO2, cosmetic grade TiO2, SiO2, silver and 1 polyacrylic acid (PAA) coated magnetic NPs].”
Corrected.
Page 2 - In the sentence of lines 86-88: “Importantly, the cytokine secretion strongly correlated with the size of the NPs in the culture media, while specific proteins in the protein corona only had moderate effects.” Reformulate the sentence to clarify the meaning of “only moderate effect of specific protein corona composition” in the cytokine secretion.
We have changed the sentence to describe our observations more clearly: »Importantly, the cytokine secretion strongly correlated with the size of the NPs in the culture media and the presence of some specific proteins in the protein corona correlated with the cytokines secretion«.
Page 2-3 – Lines 97-100: Authors should introduce references for the applications of the NPs listed below, ”c) Silver NPs as NPs widely used in various materials (citation), d) P25 (citation) and e) ANATAZ TiO2 a reference TiO2 material (citation), f) Dr. Adorable (DrA) TiO2 (citation) and g) Naturally Balmy (N.B.) TiO2 NPs used in cosmetics (citation), and h) Food-grade (FG) TiO2 that is used in food industry products (citation).”
We have added references for all the NPs as suggested.
I suggest some articles in case they are useful:
Yusuf M. (2018). Silver Nanoparticles: Synthesis and Applications. Handbook of Ecomaterials, 2343–2356. https://doi.org/10.1007/978-3-319-68255-6_16
Zhang X. F., Liu Z. G., Shen W., & Gurunathan S. (2016). Silver Nanoparticles: Synthesis, Characterization, Properties, Applications, and Therapeutic Approaches. International journal of molecular sciences, 17(9), 1534. https://doi.org/10.3390/ijms17091534
Zivic F., Grujovic N., Mitrovic S., Ahad I.U., Brabazon D. (2018). Characteristics and Applications of Silver Nanoparticles. In: , et al. Commercialization of Nanotechnologies–A Case Study Approach. Springer, Cham. https://doi.org/10.1007/978-3-319-56979-6_10
Jafari S., Mahyad B., Hashemzadeh H., Janfaza S., Gholikhani T., Tayebi L. (2020). Biomedical Applications of TiO2 Nanostructures: Recent Advances. Int J Nanomedicine, 15, 3447-3470.
https://doi.org/10.2147/IJN.S249441
Nicosia, A., Vento, F., Di Mari, G.M., D’Urso, L., Mineo, P.G. (2021). TiO2-Based Nanocomposites Thin Film Having Boosted Photocatalytic Activity for Xenobiotics Water Pollution Remediation. Nanomaterials, 11, 400. https://doi.org/10.3390/nano11020400
Jiang X., Manawan M., Feng T., Qian R., Zhao T., Zhou G., Kong F., Wang Q., Dai S., Pan J. H. (2018). Anatase and rutile in evonik aeroxide P25: Heterojunctioned or individual nanoparticles? Catalysis Today, 300, 12–17. http://dx.doi.org/10.1016/j.cattod.2017.06.010
Adawiyah J. Haider, Zainab N. Jameel, Imad H.M. Al-Hussaini. (2019). Titanium Dioxide Applications. Energy Procedia, 157, 17-29. https://doi.org/10.1016/j.egypro.2018.11.159.
Dréno B., Alexis A., Chuberre B., Marinovich M. (2019). Safety of titanium dioxide nanoparticles in cosmetics. Journal of the European Academy of Dermatology and Venereology (JEADV), 33 (7), 34–46. https://doi.org/10.1111/jdv.15943
Baranowska-Wójcik E. (2021). Factors Conditioning the Potential Effects TiO2 NPs Exposure on Human Microbiota: a Mini-Review. Biological trace element research, 199(12), 4458–4465. https://doi.org/10.1007/s12011-021-02578-5
Page 4 – Lines 140-142: In figure 2, authors should indicate the measured parameter and units of the Y-axis and include this information in the figure legend.
Corrected.
Page 6 – Lines 208-210: In figure 5, the alphabetical index of NPs is not correct. Silver NPs have been omitted, thereby, adjust the entire figure caption accordingly. Differences on incubation conditions for these NPs (25µg.ml-1 for 24h) must be included.
Thank you, we have corrected the figure legend and also added the incubation conditions for silver NPs (25 um/ml).
Page 7 – Lines 220-223: Authors should specify and add in the figure legend of Figure 6 which substance was used as positive control in the quantification of % ROS, and its concentration and incubation time.
For positive control cells were incubated with 1 mM H2O2, for 20 min, we have added this information in the figure caption.
Page 8 – Lines 232-236: In figure 7, add the meaning of N/A in the figure caption. On the other hand, data for cytokine secretion would be more robust if accompanied by statistical analysis.
We have added explanation for N/A (for silver NPs at >50 ug/ml significant cells death was present, therefore lower concentrations were used.
For the presentation of ELISA results, we present one representative repeat of three independent experiments. The absolute numbers as measured by ELISA expectedly have large variation and therefore, we have used this representation as it is commonly used. Below the figures for IL-8 with ANOVA statistic are shown.
Page 10 – Lines 310-311: The following sentence, “With ICP MS we have detected several NPs various potentially toxic ions, such as arsenic, barium and nickel.” needs to be revised and rewritten. It seems that some preposition is missing to make it sense.
Thank you, a word was missing, now is corrected: »With ICP MS we have detected several NPs various potentially toxic ions, such as arsenic, barium and nickel, however as concentrations were relatively low (≤ 1 ng ml-1) except for the silver ions in silver NPs, the impurities were probably not the major factor in observed responses.«
Page 10 and 11 – from Line 317: Dimension of Table 1 or the font size need to be adjusted, in order that all items can be fully read.
We have increased the font size.
Page 12 – Lines 373-375: The position of the three references (cites 13, 42, 86) does not seem to be quite appropriate. I suggest moving them just before the comma.
We have changed the position of these references.
Page 12 – Lines 380-381: Check the punctuation of the following sentence “Indeed. several in vivo studies demonstrated no immunogenic effect for realistic doses of TiO2 or SiO2 NPs46.”
Corrected.
Page 14 – Line 479: It would be interesting to specify and include in the methodology the dispersion protocol used by the authors, since its importance for results reproducibility is well known in nanotoxicology, and since the NPs size is one of the key properties in which the conclusions of the study is based on.
We agree that this is an important issue. We have included more details of the dispersion protocol in the Method Section 4.2 Nanoparticles characterisation:« NPs original stocks were first vortexed and dispersed before the measurements in either distilled water or cell culture media at the following concentrations: 0,05 w/w% for PAA, SiO2, Silver NPs and 0,02 w/w% for all TiO2 NPs, NPs suspension were re-suspended several times with the pipet.«
In the protocols for cell incubation with the NPs (Section 4.7) we have also added: »NPs original stocks were first vortexed and dispersed. After NPs were added to the cell media suspensions were re-suspended several times with the pipet«.
Page 17 – Conclusions: In general, this section is quite long. The authors should avoid repeating the description of results in some paragraphs, and only expose the conclusions drawn from the study outcomes.
We have significantly shortened the conclusion section and omitted some sentences.

Reviewer 2 Report
Generally, the results of the study are interesting and the methods used are appropriate, however in order to give consistency to the article additional modifications of the manuscript are required.
In order to better support the results of the study the homology of the primary structure of human serum proteins and bovine fetal serum proteins should be analyzed due to the human origin of THP-1 cell line in relation with the NP protein corona (e.g., in a table).
Please provide in the supplementary information section the SDS-PAGE electrophoregrams (gel images) and the LC-ESI-MS / MS spectra for a better understanding of the results.
Figure 3 is difficult to understand; please provide more information in the Figure’s Legend about the spectral counts and the meaning of 0-250 values that appear in the histogram. The graphical representation of cell viability (Figure 4) is not consistent with the method of determination (4.7). We recommend the transformation of Figures 8 and 9 into tables, due to the large differences between the values represented in (ng/mL) of secreted IL-1b and IL-8 interleukins according the treatments applied.
Author Response
We thank the reviewer for positive comments and suggestions.
In order to better support the results of the study the homology of the primary structure of human serum proteins and bovine fetal serum proteins should be analyzed due to the human origin of THP-1 cell line in relation with the NP protein corona (e.g., in a table).
We agree with the reviewer that this is an important issue. We have included in the Supplementary file Table S1 additional data in order to compare the similarity of the primary structures of the identified bovine proteins in the corona of NPs with their human orthologues. The results are listed in a new column in Table S1. Considering the high sequence similarity and the similar abundance of the major proteins in bovine and human serum, we can assume that the composition of the hard protein corona of the NPs will be similar in both cases. Furthermore, some of the identified corona proteins that showed possible correlations with observed toxicological endpoints showed very high sequence identity/similarity: ANT3 (87/92), HSP90A (99/100) and TSP1 (97/98). However, we agree that our results with bovine serum cannot simply be extrapolated to human serum, as we have already written in the Discussion section (lanes 451-454):
“In general, our results suggest that the composition of protein corona and presence of specific proteins might be an important factor for physiological response, but to make definite conclusions a broader study with a larger number of NPs and human serum is needed.”
Please provide in the supplementary information section the SDS-PAGE electrophoregrams (gel images) and the LC-ESI-MS / MS spectra for a better understanding of the results.
We have added SDS-PAGE representative gel images (Supplementary information Figure S3)
Figure S3. Analysis of NPs protein corona using SDS-PAGE, polyacrylic acid coated NPs (PAA), silver NPs (Ag), SiO2, TiO2 food-grade (FG), TiO2 P25 (P25), TiO2 N.B. (N.B.), TiO2 21nm (21 nm), TiO2 ANATAZ (Anat) and TiO2 DrA NPs. Please note that additional NPs denoted as TiO2 21nm were added as a pilot test of another bach of P25 TiO2. The bands A-G (mobbed red) were excised and analysed by MS (see Table S2). M denotes lanes loaded with molecular mass standards.
and all LC-MS raw data (See supplementary file »Table S2-MS.excl«):
Table S2. Identification of nanoparticle corona proteins separated by SDS-PAGE (Figure S3) using the ESI-MS/MS. Three independent experiments were performed. Abbreviation: m, oxidised methionine.
Figure 3 is difficult to understand; please provide more information in the Figure’s Legend about the spectral counts and the meaning of 0-250 values that appear in the histogram.
In Figure 3 we show heatmap of the exclusive spectral counts obtained from MS – average of the three independet measurements. Since the highest average absolute number (in all NPs sample and for all proteins) of detected spectral counts was 250 (albumin in corona of TiO2 DrA) the scale in the heatmap is between 0 (none detected) and 250 (the maximum value).
We agree this was not explained enough and we have added additional information in the figure legend: „Heatmap of protein abundance (spectral counts obtained from MS) and B) protein abundance, the average values of three independent repeats are presented (see also Figures S4, S5).“
The graphical representation of cell viability (Figure 4) is not consistent with the method of determination (4.7).
We agree that the description in 4.7 was not complete – we have changed this part of the text:
The percentage of cell viability (% Viability) was determined as the ratio between the number of viable cells in each sample (NS = Hoechst positive-PI positive) and the number of all cells in the non-treated control (N0): %Viability = 100 × NS/N0. The number of dead cells (%dead) was calculated as %Dead = 100 x Number PI positive/Number Hoechst positive).
We recommend the transformation of Figures 8 and 9 into tables, due to the large differences between the values represented in (pg/mL) of secreted IL-1b and IL-8 interleukins according the treatments applied.
Figure 8 shows only IL1-beta. Within each figure, the values fall within an expected range. We feel that the graphical representation is clearer and have left the figures as they are, as it would be difficult to see the dose-dependent trends and differences between NPs in the table.
In Figure 9, the scale of the y-axis is different because a completely different range of cytokine secretion is expected for different cytokines (different mechanism, priming is also required for IL -1beta).

Reviewer 3 Report
The paper named “ The relevance of physico-chemical properties and protein corona for evaluation of nanoparticles immunotoxicity – in vitro correlation analysis on THP-1 macrophages” is an interesting work were author provides new insights towards better understanding of the relationships between alongside physiochemical properties (PCP), protein corona and inflammatory response of macrophages for different engineered NPs. Therefore in this paper author correlated PCP and protein corona composition of NPs to THP-1 macrophages response, focusing on selected toxicological endpoints: cell viability, reactive oxygen species [ROS], cytokine secretion.
Only minor points are required:
1)In line 106 instead of figure 1 the authors must put figure 1a since the information indicated is in that figure
2) In paragraph between lines 115-117 a reference to figure 1B is missing
3) In figure 2 no data are represented from Si however author says that “In general, Si was found in higher quantities than other elements due to sample preparation in glass bottles and was therefore excluded from the analysis. Can author explain this point?
4) In the same figure (figure 2) where author analyzed the impurities author do not shown data about Mg only about Mn however in the text they say that Mg is detected in some NPs. Please check
5) Point 2.3 is made in culture media after its exposition to cells or in cells? This mean that the protein corona depends in the media that the NPs are analyzed, in this case author used culture media therefore the proteins that join to the NPs must be due to the media component. Why author, if want to see the immune system, do not use human serum? It is necessary that author explain this point because in point 4.4 only media incubation in order to make the NPs protein corona is explains.
6) In heat map the replicates must be added
7) What is the negative control in protein corona analysis?
8) To see the cell viability author make a Hoechst/PI differential staining assay have author make other analysis as FACS analysis to see the different cellular phase after NPs exposition?
9) Please clarify the means of PC in figure 6, 7 and 8.
10) Why author used different concentration in silver NPs with respect to the others in figure 6, 7 and 8?
12) How it is possible that in Silver NPs there is not IL-1b at 100mg/ml
11) In figure 9 what means PC 24h and PC 72h?. In the same figure the resolution of IL8 in silver NPs are not significative? They must put in the graph the signification value.
13) In the discussion in line 308 a reference to fig 2 is missing
Author Response
We thank the reviewer for his valuable suggestions.
Only minor points are required:
1)In line 106 instead of figure 1 the authors must put figure 1a since the information indicated is in that figure
Corrected.
2) In paragraph between lines 115-117 a reference to figure 1B is missing
Corrected.
3) In figure 2 no data are represented from Si however author says that “In general, Si was found in higher quantities than other elements due to sample preparation in glass bottles and was therefore excluded from the analysis. Can author explain this point?
In Figure 2 we have omitted Mg and Si as we suspected that they may be an artefact of the preparation. However, they were measured above the values of the blank samples, so we have now added the elements Mg and Si as well.
We have now replaced this graph with the graph including also elements Mg and Si. We thank the reviewer for this observation.
4) In the same figure (figure 2) where author analyzed the impurities author do not shown data about Mg only about Mn however in the text they say that Mg is detected in some NPs. Please check
As described above – corrected.
5) Point 2.3 is made in culture media after its exposition to cells or in cells? This mean that the protein corona depends in the media that the NPs are analyzed, in this case author used culture media therefore the proteins that join to the NPs must be due to the media component. Why author, if want to see the immune system, do not use human serum? It is necessary that author explain this point because in point 4.4 only media incubation in order to make the NPs protein corona is explains.
We agree with the reviewer that this is important, and we had considered this point. Most in vitro and in vivo studies with NPs are conducted with bovine serum, mainly because it is available and traditionally used in many tests, but also for economic reasons. Also, all primary optimization of macrophages differentiation was performed in this medium RPMI + 10% FBS.
It is known that the protein corona is a dynamic system and would change after incubation of NPs with cells. Therefore, we decided to use the same serum (FBS) for the formation of the NPs corona as in the media for THP-1 cell growth, viability and ELISA experiments.
However, we agree that it important to analyse NPs also in human serum (which is stated in the Discussion) and that it is necessary to compare the homology of the human and bovine proteins which is now presented in Supplementary file Table S1. Considering the high sequence similarity and the similar abundance of the major proteins in bovine and human serum, we can assume that the composition of the hard protein corona of the NPs will be similar in both cases. Furthermore, some of the identified corona proteins that showed possible correlations with observed toxicological endpoints showed very high sequence identity/similarity: ANT3 (87/92), HSP90A (99/100) and TSP1 (97/98). But we agree that the results with bovine serum cannot simply be extrapolated to human serum, as we have already written in the Discussion section (lanes 451-454).
6) In heat map the replicates must be added
In the supplementary information file we have added Figure S4 with the heatmaps together with the spectral counts of the three biological replicates.
7) What is the negative control in protein corona analysis?
We used the same procedure to obtain the negative control as in Strojan et al., PLOS ONE 2017; For the negative control, the procedure (4.4. NPs corona preparation) was carried out similarly to that for the formation of the NPs corona, except that the same volume of PBS was incubated with the serum instead of the NP suspension. Further preparation followed the protocol described in sections 4.4 and 4.5 (sodium dodecyl sulphate-polyacrylamide gel electrophoresis (SDS-PAGE) and mass spectrometry (MS).
8) To see the cell viability author make a Hoechst/PI differential staining assay have author make other analysis as FACS analysis to see the different cellular phase after NPs exposition?
We did not perform an analysis with flow cytometry because we had previously observed that with the Hoechst/ PI differential staining the cell count and the number of dead cells are shown together, from which the possible effects on proliferation can also be read. We agree that it would be interesting to determine the effects on the distribution of cells in relation to the cell cycle, but we did not perform this analysis.
9) Please clarify the means of PC in figure 6, 7 and 8.
We have added in the figure legends the data for the positive controls.
Figure 6: For the positive control (PC) cells were incubated with 1 mM H2O2 for 20 min.
In Figure 7: Cells exposed to 100 ng ml-1 LPS for 24 h were used as a positive control.
Figure 8: 200 µg ml-1 crystalline SiO2 NPs was used as a positive control.
10) Why author used different concentration in silver NPs with respect to the others in figure 6, 7 and 8?
With silver NPs, excessive cell death already occurred at 50 µg ml-1 (only 20% living cells). Therefore, we had used lower concentrations of Ag NPs. We had added in the figure legends: "Please note that lower concentrations were used for silver NPs due to excessive cell death."
12) How it is possible that in Silver NPs there is not IL-1b at 100mg/ml
We had not used 100 um/ml Ag as all cells were dead. Therefore the notation N/A.
11) In figure 9 what means PC 24h and PC 72h?. In the same figure the resolution of IL8 in silver NPs are not significative? They must put in the graph the signification value.
We have added in the figure legends more details of the PC: »100 ng ml-1 LPS for IL-8 or 200 µg ml-1 crystalline SiO2 NPs for IL-1β were used as a positive control with 24 h incubation or 72 h incubation, for PC 24 h and PC 72 h, respectively.«
13) In the discussion in line 308 a reference to fig 2 is missing
Corrected.
